# An Efficient 2.5D Shadow Detection Algorithm for Urban Planning and Design Using a Tensor Based Approach

**Sukriti Bhattacharya** [1],*[ID]**, Christian Braun** [2][ID] **and Ulrich Leopold** [2][ID]

1   Department for IT for Innovative Services, Luxembourg Institute of Science and Technology (LIST),
    L-4362 Esch-sur-Alzette, Luxembourg
2   Department for Environmental Research and Innovation, Luxembourg Institute of Science and Technology
    (LIST), L-4362 Esch-sur-Alzette, Luxembourg; christian.braun@list.lu (C.B.); ulrich.leopold@list.lu (U.L.)
*   Correspondence: sukriti.bhattacharya@list.lu

**Abstract:** Urbanization is leading us to a more chaotic state where healthy living becomes a prime concern. The high-rise buildings influence the urban setting with a high shadow rate on surroundings that can have no positive impact on the general neighborhood. Nevertheless, shadows are the main factor of defeatist virtual settings, they are expensive to render in real-time. This paper investigates how the amount of sunlight varies by season and how seasons can indicate the time of year to understand how shadows vary in length at different times of the day and how they change over the seasons. We propose a novel efficient (fast and scalable) algorithm to calculate a 2.5D cast-shadow map from a given LiDAR-derived Digital Surface Model (DSM). We present a proof-of-concept demonstration to examine the technical practicability of the introduced algorithm. Tensor-based techniques such as singular value decomposition, tensor unfolding are examined and deployed to represent the multidimensional data. The proposed method exploits horizon mapping ideas and extends the method to a modern graphics algorithm (Bresenham's line drawing algorithm) to account for the DSM's underlying surface geometry. A proof-of-concept is developed utilizing Python's TensorFlow library, exploring data flow graphs and the tensor data structure. The heavy computer graphics algorithm used in this paper is parallelized using PySpark. Results explicate significant enhancements in overall performance while preserving accuracy at negligible variations.

**Keywords:** tensor; TensorFlow; Digital Surface Model; Bresenham's Algorithm; cast shadow

## 1. Introduction

Solar Access has become an essential concern for urban planning and building design. Access to sunlight is a necessary element of a healthy individual thermic relief for inhabitable buildings. Direct sunlight is essential in architecture for psychic health reasons and reduces electric lighting and energy saving. Therefore, spatial analysis [1] in an urban environment often requires evaluating shadow points given a specific date and time, precisely associated with the solar position and shared representation of spatial obstructions such as buildings. Several zonal authorities now require light concerns to be spoken as part of the marking. Shadow studies explain the value of development concerning sun and daylight access to the neighboring settings, that includes encompassing buildings. The following scenarios make shadow calculation a crucial aspect of urban planning,

1.  Calculating the amount of time a given roof or facade is shaded, to determine the utility of installing photovoltaic cells for electricity production [2].
2.  Calculating shadow footprint on vegetated areas, to determine the expected influence of a tall new building on the surrounding microclimate [3].
3.  Daylight access; Shadow impact calculation to maximize the use of private residential amenity spaces during spring, summer, and fall (Standards For Shadow Studies— http://www.mississauga.ca, accessed on 12 June 2021; Shadow Impact Analysis— https://www.cityofdavis.org, accessed on 25 November 2020).

The Sun is the most notable light source in the solar system, which transmits light from a single point in space; only one ray of direct light can hit a spot on the surface. Therefore, the allied intensity level can only be varied, depending on whether an obstruction prevents the ray. According to the principle of formation, there are two shadow types, cast-shadow, and self-shadow. The cast-shadow also referred to as hard shadow or umbra, is formulated by the objects' projection in the light source's direction. On the other hand, the self-shadow relates to the part of the object that is not illuminated, i.e., the building's façade on the opposite side of the light source.

The main idea of horizon mapping is to compute shadows for bump-mapped surfaces by precomputing each height-field visibility. The visibility is determined by the slope, which represents the highest elevation spot seen from a particular viewpoint. Max [4] proposed horizon mapping to overcome missing shadows on bump mapped surfaces. The bump mapping is a basic shading technique for rendering bumps on flat textures proposed by Blinn [5]. Cook introduced the displacement mapping to represent bumps [6]. The shadow map method [7,8] and shadow volume method [9] showed displacement mapping could be combined with rendering methods for geometry-based shadows. Becker and Max [10] showed that various bump mapping algorithms could be blended, including displacement mapping. Later, Wang et al. developed a view-dependent displacement mapping algorithm [11]. Onoue et al. took surface curvature into account to compute bump map shadows [12]. In late 1968 by A. Appel [13] introduced ray tracing for hard shadow. The principle is straightforward; a shadow ray casts from the intersection point to the light source. If the ray intersects an object between its origin and the light source, it is in shadow; otherwise. The raster-based solutions on the other hand, are widely implemented in softwares and tools [14,15]. For example, `r.sun` command [16] in GRASS GIS [17], the UMEP plugin [18] for QGIS [19], `insol` (https://CRAN.R-project.org/package=insol, accessed on 22 March 2019) package [20] and `shadow` (https://CRAN.R-project.org/package=shadow, accessed on 22 March 2019) package [21] in R, are few widely available open-source solutions. The proprietary software ArcGIS, provides raster-based shadow calculations using the Solar Analyst extension [22].

In one of our recent works [23], we proposed a 2.5D shadow detection algorithm in a similar context. The term 2.5D is used in computer graphics to describe the process of creating the illusion of a three-dimensional scene without the use of complete 3D rendering. 2.5D projections have been helpful in understanding visual-cognitive spatial representations or 3D visualization in geographic visualization (GVIS) [24]. The main limitations of the proposed algorithm are as follows,

- The algorithm was limited to the buildings only. The base height of the buildings was calculated using `3Dfier` (https://github.com/tudelft3d/3dfier, accessed on 22 March 2019) developed by Delft University. `3Dfier` uses the LOD1 model [25] in which blocks represent buildings; Therefore, it overlooks accurate building height.
- the shadow point calculation algorithm was not parallelized. Hence time-consuming.

This paper proposes a novel scalable 2.5D real-time hard shadow detection algorithm to overcome the shortcomings. First, the algorithm operates on a LiDAR-derived Digital Surface Model (DSM) at a sufficiently high resolution for a given urban context. DSM represents the mean sea level (MSL) elevations of trees' reflective surfaces, buildings, and other features elevated above the bare earth. Therefore, the present algorithm is not dependent on `3Dfier` and limited to the LOD1 model. Secondly, from an implementation point of view, tensor-based techniques such as singular value decomposition, tensor unfolding are examined and deployed to represent the multidimensional data [26]. The proposed method exploits horizon mapping ideas and extends the technique to a modern graphics algorithm (Bresenham's line drawing algorithm) to account for the DSM's underlying surface geometry. The proof of the concept is developed in Python using Tensorflow [27], an open-source software library developed by the Google Brain Team using data flow graphs and the tensor data structure. The heavy computer graphics algorithm used in this

paper is parallelized using PySpark [28,29]. Results obtained from the stated work were analyzed to exhibit high scalability levels in time and accuracy.

## 2. Methodology

### 2.1. Shadow Geometry

Solar orientation is the means of aiming something at the sun. Solar technologies like a solar panel or a sun oven receive the highest energy amount when oriented at a right angle (90°) towards the sun. Standing at the exact center of the skydome, where the sun is somewhere at the inside surface of the dome, then the sun's position on the skydome is a combination of two angles, as shown in Figure 1.

The solar altitude (elevation) angle is the angle formed by the sun's rays and a horizontal plane. On the other hand, solar azimuth angle indicates the Sun's compass position. Figure 1 depicts solar azimuth angle measured from due north to equal 180° at solar noon. As shown in Figure 2, the position of the Sun in the sky can be defined by different angles from any point (P) on the earth's surface, comprising longitude and latitude, pinpoints the location P on the earth and a set of angles related to the sun's position regarding a precise location on the earth on a given date and time. These angles are declination angle, elevation/altitude angle($\alpha$), zenith angle($\beta$), azimuth angle($\psi$), and hour angle($\omega$).

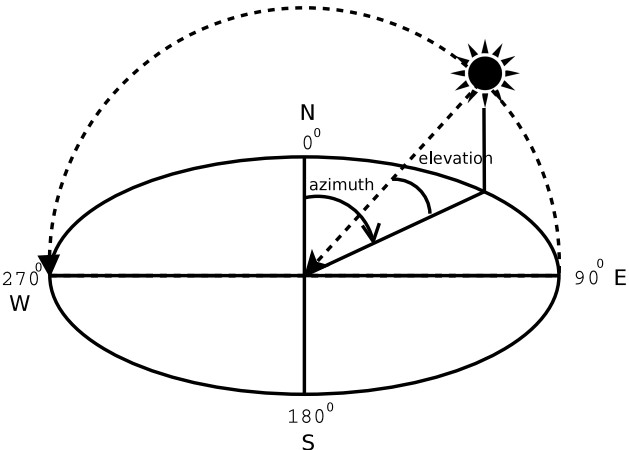

**Figure 1.** The Sun Position Diagram.

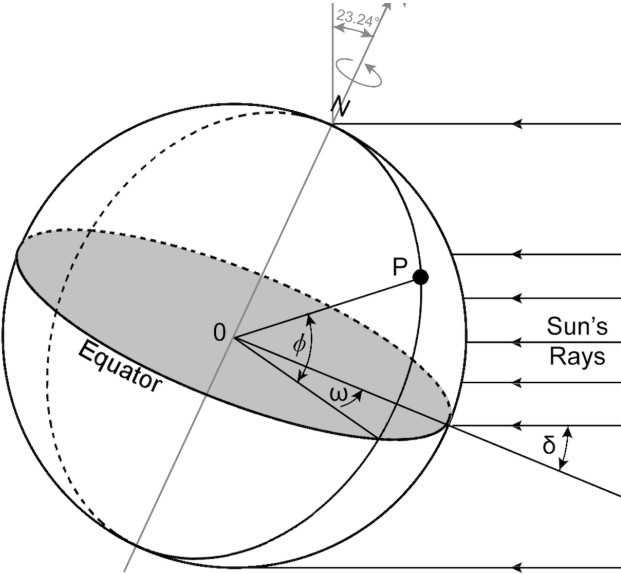

**Figure 2.** Solar Angles.

For a given date(d) and a given period of time intervals in minute($\tau$) , Algorithm 1 [23] computes the sun position from sunrise to sunset for a specific latitude($\phi$). $\Delta$ in line 3 is the day angle [16] and n is the day number, which varies from 1 on 1 January to 365 (366) 31 December. The hour angle is the angular displacement of the sun east or west of the local meridian due to the rotation of the earth around its axis 15° per hour; morning negative, afternoon positive and during solar noon zero. The hour angle for sunrise ($\omega_{sr}$) and sunset ($\omega_{ss}$) is explained in line 5 and line 6, respectively. Line 7 converts the time interval $\tau$ to angle in radian $\tau_\omega$. The loop at line 8 calculates the hour angles, $I_\omega$ from sunrise to sunset. The solar elevation angle ($\alpha$), the complement of the zenith angle is the angle between the horizontal plane and the line to the sun as depicted in Figure 2. It is zero both at sunrise and sunset and 90° when the sun is directly overhead, which occurs at the equator on spring and fall equinox. Line 12 calculates the elevation angle at hour angle $\omega_i$. The solar azimuth angle calculated through line 13 to 17 is the angular distance of the sun's projection on the horizontal plane at a place on earth from a reference direction. Different authors use different conventions for calculating the solar azimuth angle. In this paper, the azimuth angle is measured from the north clockwise from zero to 360°.

---

**Algorithm 1** Sun Position Calculation from Sunrise to Sunset

---

1: **procedure** SUNPOS($\phi$, $\tau$, d )
2:     n = dayofyear(d)
3:     $\Delta = (2.0 \times \pi \times \text{n})/365.25$
4:     $\delta = \sin^{-1}(0.3978 \times \sin(\Delta - 1.4 + 0.0355 \times sin(\Delta - 0.0489)))$
5:     $\omega_{ss} = \cos^{-1}(-\tan(\phi) \times \tan(\delta))$
6:     $\omega_{sr} = -\omega_{ss}$
7:     $\tau_\omega = \tau \times 0.261799$
8:     **for** ($\text{i} = \omega_{sr}$; $\omega_{ss} \leq \omega_{sr}$; $\text{i} = \text{i} + \tau_\omega$) **do**
9:         $I_\omega = \text{i}$
10:     **end for**
11:     **for each** $\omega_i \in I_\omega$ **do**
12:         $\alpha[\text{i}] = \sin^{-1}(\sin\delta \times \sin\phi + \cos\delta \times \cos\omega_i \times \cos\phi)$
13:         $\psi' = \cos^{-1}\left((\sin\delta \times \cos\phi - \cos\delta \times \cos\omega_i \times \cos\phi)/cos\alpha\right)$
14:         **if** ($\text{i}_\omega \geq 0$) **then**
15:             $\psi' = 360° - \psi'$
16:         **end if**
17:         $\psi[\text{i}] = \psi'$
18:         i++
19:     **end for**
20:     **Return** $\alpha, \psi$
21: **end procedure**

---

In Figure 3, $\triangle$OPQ form a right angle triangle considering the fencepost OQ of height, h drove perpendicularly into the ground. Now the question is, where would the shadow fall? However, it is very easy to figure it out by drawing a line from the sun across the top of the fencepost to the ground.

In this case, the length of the shadow |OP| can be calculated using the acute angle between the horizon and the line to the sun, $\alpha$ ($\angle$OPQ) in Equation (1). The angle $\alpha$ is commonly known as the elevation angle or altitude angle.

$$\tan(\alpha) = \frac{h}{|OP|} \Rightarrow |OP| = \frac{h}{\tan(\alpha)} \text{ meter.} \tag{1}$$

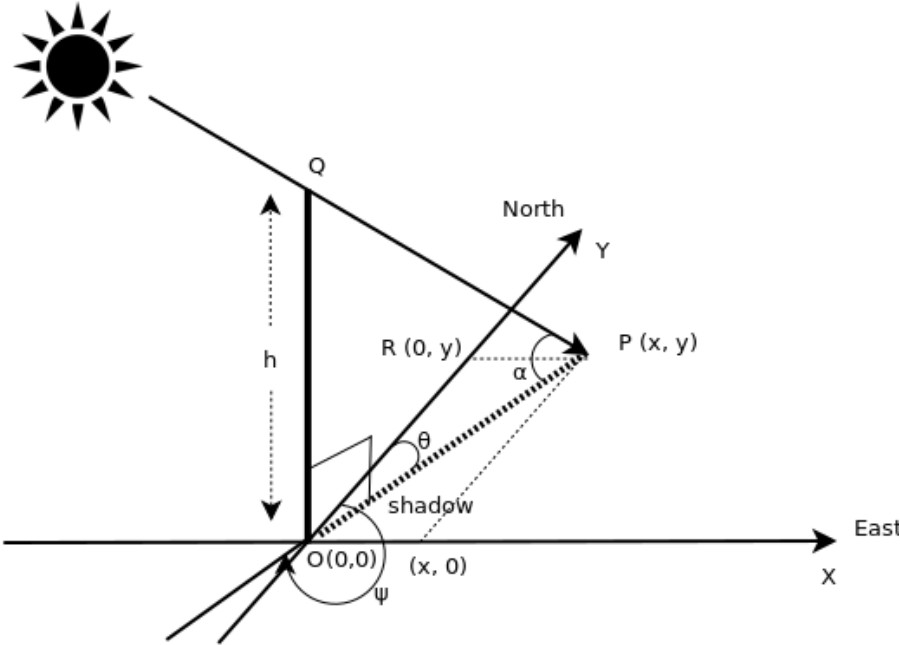

**Figure 3.** Shadow Calculation.

Our goal is to calculate the tip of the shadow located at P(x, y), i.e., x meter East and y meter North of the base of the fencepost. Using right triangle trigonometry on △ORP, we got $\theta = \psi - 180°$,

$$\sin\theta = \frac{x}{|OP|} \Rightarrow x = |OP| \times \sin\theta \tag{2}$$

In Equation (2), $\psi$ is a horizontal angle measured clockwise from a north base line, commonly known as solar azimuth angle. Substituting |OP| from Equation (1) to Equation (2) yields x in Equation (3).

$$x = \frac{h}{\tan\alpha} \times \sin\theta = -\frac{h \sin\psi}{\tan\alpha} \tag{3}$$

The same way, y can be calculated by Equations (4) and (5).

$$\cos\theta = \frac{y}{|OP|} \Rightarrow y = |OP| \times \cos\theta \tag{4}$$

$$y = \frac{h}{\tan\alpha} \times \cos\theta = -\frac{h \cos\psi}{\tan\alpha} \tag{5}$$

Therefore, from Equations (3) and (5) the shadow tip coordinate can be observed as

$$P = \left( -\frac{h \sin\psi}{\tan\alpha}, -\frac{h \cos\psi}{\tan\alpha} \right) \tag{6}$$

From Equations (2) and (5) the shadow tip coordinate can be observed as

$$P = \left( |OP| \times \sin\theta, |OP| \times \cos\theta \right) \tag{7}$$

### 2.2. Shadow Condition on DSM

The digital surface model (DSM) represents the top of the earth's surface -anything that sits on the earth, including trees, buildings. hills etc. Let us consider two points A and P on earth surface as shown in Figure 4. The height (from sea level) of A and P are $h_1$ (AB) and $h_2$ (PQ), respectively. △PKA is right angle triangle, where ∠PKA = 90°. The angle $\beta$

($\angle$PAK) represents the horizon angle at A. Using right angle triangle trigonometry $\beta$ can be calculated as,

$$\beta = \tan^{-1}\left(\frac{h_2 - h_1}{AK}\right) \tag{8}$$

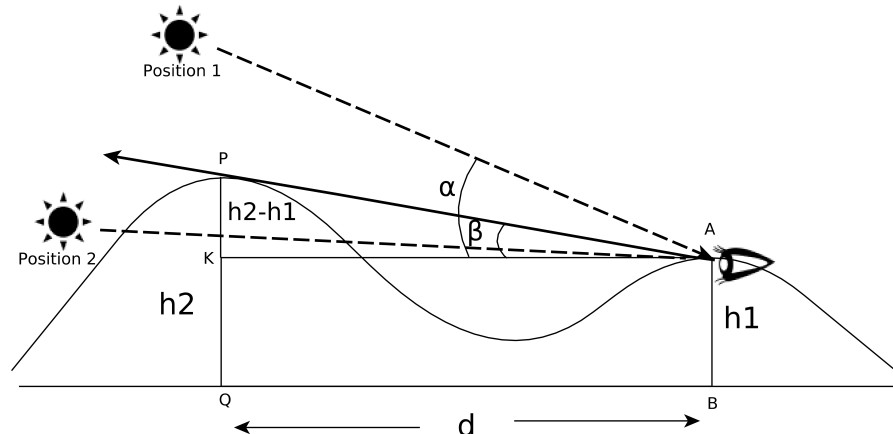

**Figure 4.** Shadow Calculation on Bumped Surface.

Using shadow mapping technique we can approximate the shadow by testing whether point A(x,y) is visible from the light source (the sun),

$$A(x,y) = \begin{cases} 1, & \text{if } \alpha < \beta \implies \text{shadow} \\ 0, & \text{otherwise} \end{cases} \tag{9}$$

The visibility condition is governed by the elevation angle $\alpha$. As shown in Figure 4, the sun (at Position 2) is not visible from point A as elevation angle ($\alpha$) is less than the horizon angle ($\beta$), yiled A in shadow. Likewise, when the sun is at Position 1 point A is not observed as shadow point.

### 2.3. 2.5D Shadow Calculation Algorithm on DSM

The main shadow calculation process shadowpoints is stated formally in Algorithm 3. shadowpoints takes the following inputs,

1. R$\langle$n,s,e,w$\rangle$: Given region specified by north- south-east-west coordinates $\langle$n,s,e,w$\rangle$.
2. date: Given date.
3. time: Given time.
4. D: For any location (x,y) within R, D is the radius of the neighborhood centering (x,y).
5. DSM: Digital Surface Model of R.

As output, shadowpoints estimates a set of points, shadow within region R that are in shadow. The latitude values associated to R$\langle$n,s,e,w$\rangle$ are extracted using extractLat function in line 2. The sun position at a given date and time is determined by sunpos_t function in line 3. sunpos_t extracts the exact sun position by means of solar azimuth angle ($\psi$) and altitude angle ($\alpha$) at a given time, time from the sun-path calculated by sunpos algorithm(Algorithm 1), explained in Section 2.1. In line 4, shadowpoints calls shadowtop (Algorithm 2). shadowtop returns the actual coordinates (XY) and associated shadow-top coordinates (XY$_\top$) of each raster cell in R. The function coordinates in line 3 of shadowtop gives the (x,y) coordinates of each cell in R. Associated shadow-top coordinates are generated (from line 7 to 10 in Algorithm 2) from the displacements ($d_x$, $d_y$) (from line 4 to 6 in Algorithm 2) for a given distance D. For any location (x,y) within R, D is the radius of the neighborhood centering (x,y). The underlying shadow geometry is explained in Section 2.1.

---

**Algorithm 2** Calculate Shadow Top

---

1: **procedure** SHADOWTOP(R⟨n,s,e,w⟩, date, time, D)
2:     $\psi, \alpha$ = sunpos_t(SUNPOS($\phi, \tau$, date), time)
3:     XY = coordinates($R$⟨n,s,e,w⟩)
4:     $\theta = \psi - 180°$
5:     $d_x$ = D * $\sin \theta$
6:     $d_y$ = D * $\cos \theta$
7:     **for** each $(x,y) \in$ XY **do**
8:         $x_\top$ = $x - d_x$
9:         $y_\top$ = $x - d_y$
10:         $XY_\top$ = construct$_\top$($x_\top, y_\top$)
11:     **end for**
12:     **Return** XY, $XY_\top$
13: **end procedure**

---

Coming back to the main shadow calculation algorithm, the next step is to find out the points that lay in the path from each origin $(x,y) \in$ XY to its associated shadow-top coordinates $(x_\top, y_\top) \in XY_\top$. We accomplished this task using Bresenham's line drawing algorithm [30] used in computer graphics, explained in next section (Section 2.1). Lines 6 to 23 in shadowpoints iterate for each point pair belongs to XY and $XY_\top$, respectively.

---

**Algorithm 3** Calculate Shadow On Digital Surface Model

---

1: **procedure** SHADOWPOINTS(R⟨n,s,e,w⟩, date, time, D, DSM)
2:     $\phi$ = extractLat(R⟨n,s,e,w⟩)
3:     $\psi, \alpha$ = sunpos_t(sunpos($\phi, \tau$, date), time)
4:     XY, $XY_\top$ = shadowtop(R⟨n,s,e,w⟩, $\phi$, date, time, D)
5:     shadow = $\{\varnothing\}$
6:     **for** each $(x,y) \in$ XY & $(x_\top, y_\top) \in XY_\top$ **do**
7:         line$_{PTS}$ = BRESENHAM($(x,y)$, $(x_\top, y_\top)$)
8:         source = $(x_0, y_0) \in$ line$_{PTS}$
9:         rest = line$_{PTS}$ − $\{(x_0, y_0)\}$
10:         **for** each $(x',y') \in$ rest **do**
11:             $\beta_{(x',y')} = \tan^{-1}\left(\frac{\text{DSM}_{(x_0,y_0)} - \text{DSM}_{(x',y')}}{\sqrt{(x_0-x')^2 + (y_0-y')^2}}\right)$
12:             **if** $\beta_{(x',y')} \geq \alpha_{(x',y')}$ **then**
13:                 Is_shadow$_{(x',y')}$ = 1
14:             **else**
15:                 Is_shadow$_{(x',y')}$ = 0
16:             **end if**
17:         **end for**
18:         shadow$_{PTS}$ = $\{\varnothing\}$
19:         **while** Is_shadow$_{(x',y')} \neq 1$ **do**
20:             shadow$_{PTS}$ = shadow$_{PTS}$ $\cup$ $\{(x',y')\}$
21:         **end while**
22:         shadow = shadow $\cup$ {shadow$_{PTS}$}
23:     **end for**
24:     **Return** shadow
25: **end procedure**

---

line$_{PTS}$ $\subseteq$ XY in line 7 contains all the points that are supposed to be in shadow. source is the origin coordinate of that specific line and rest $\subset$ line$_{PTS}$ is the set of the points expect the source. In line 10 the horizon angle $\beta$ at each point in rest is calculated and the shadow condition is observed to construct the shadow map Is_shadow, as described in Equation (9) in Section 2.1. source is considered as the reference point for each point in rest. The algorithm traverse Is_shadow towards the shadow-top (line 19) from source to

encounter the first shadow point (first 1) to occur. For example, the locations `Is_shadow[0]` to `Is_shadow[i]` will be in shadow if `Is_shadow[i]` contains 1. Therefore, the associated coordinates say, $(x_0, y_0)$ to $(x_i, y_i)$ will be added to the set `shadow`$_{\text{PTS}}$ (line 19). Finally, the set `shadow` contains the points that are in shadow within the region `R`.

## 3. Proof-of-Concept

We developed a proof-of-concept to validate the shadow calculation algorithm proposed in the previous section. The proof-of-concept implements a tensor based framework for spatio-temporal raster data processing proposed in [31]. Figure 5 shows the principal steps of the proposed methodology. The system can be broken down into three main modules,

1.  Tensor data-frame construction.
2.  Sun position and shadow-top coordinate generation (using Tensorflow).
3.  Shadow points detection (using PySpark).

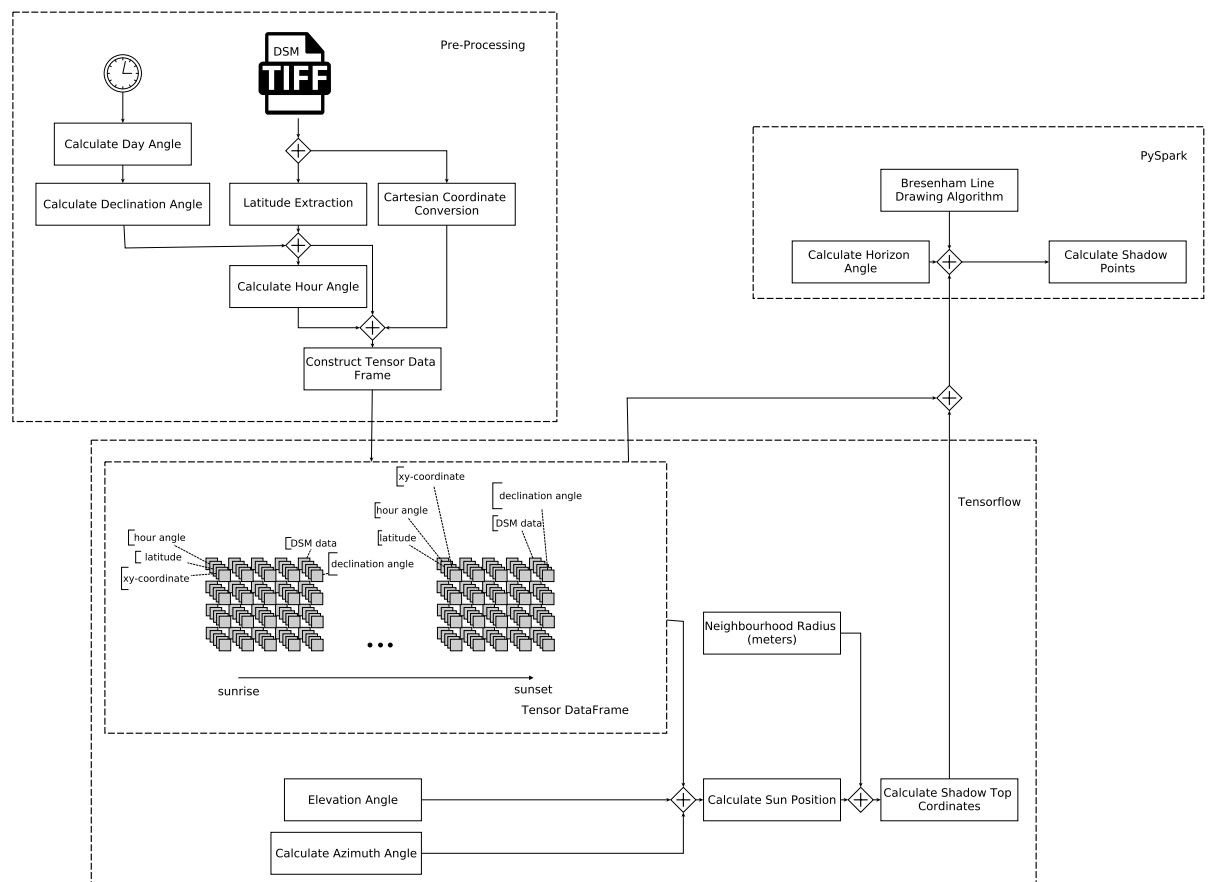

**Figure 5.** Block Diagram of Shadow Calculation Approach.

### 3.1. Tensor Data-Frame Construction

In raster data models, spatial information can be represented as a raster map consisting of a grid of cells or a matrix. Each cell stores a value that represents a specific area. The resolution of the map defines the size of the area. Matrix rows represent the x-axis and columns the y-axis in a cartesian plane. Stored values in each cell represent continuous data that can be altitude or temperature, or categorical data, etc. Formally described, a raster $R$ with $m$, $I = \{i_1, i_2, \ldots, i_m\}$ rows and $n$, $J = \{j_1, j_2, \ldots, j_n\}$ columns is a set of fixed regularly distributed locations in space such that adjacent locations hold a constant distance. Observations are recorded on a fixed set of timestamps $\mathcal{T} = \{c_1, c_2, \ldots, c_t\}$ for

every location. That can again be regularly spaced with equal delays between consecutive measurements, as illustrated in Figure 6.

Tensors are mathematical objects—basic generalizations of vectors and matrices to potentially higher dimensions. A tensor is described by its order, a unit of dimensionality, its shape, the size of each dimension, and a static type assigned to the tensor's elements. The general convention to denote an order N tensor is by upper case bold Euler script letters $\mathcal{X} \in \mathbb{R}^{I_1 \times I_2 \times \cdots \times I_N}$. Similarly, A tensor slice is a subfield that can be extracted by fixing all but two tensor indices.

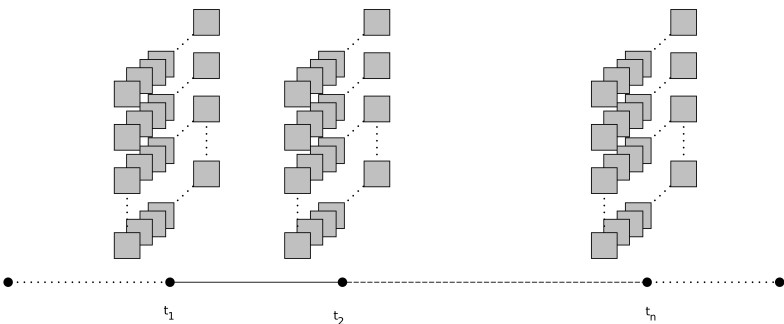

**Figure 6.** Observations for equal time intervals.

A tensor data-cube is constructed for each timestamp shown in Figure 7. Where, the Cartesian product of $R \times T$ results in the spatio-temporal tensor grid $\mathcal{X} \in \mathbb{R}^{(t \times m \times n)}$, where every cell on the tensor $\mathcal{X}$, $(c, i, j)$, has a distinct measurement value.

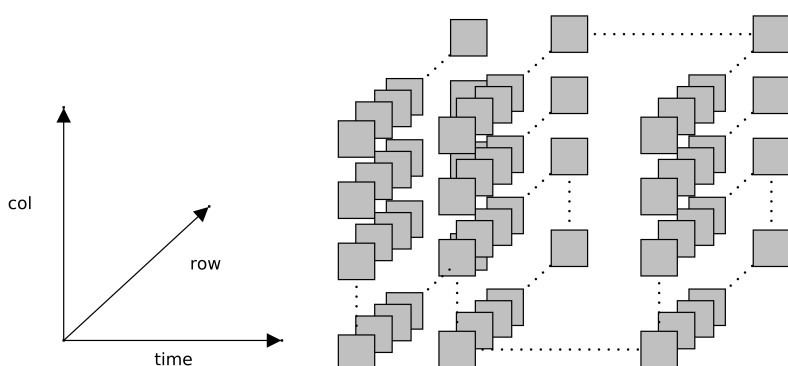

**Figure 7.** Observations stored in a single 3-D tensor.

The proof-of-concept system accepts a `GeoTIFF` file as input containing the DSM values of a city. For a selected sub-region `R⟨n,s,e,w⟩`, associated DSM, latitude, the Cartesian coordinates values are extracted. For a given date and time hour angle and declination angle are also computed. These values eventually form a tensor data-cube for each timestamp. The shape of each tensor data-cube at a specific timestamp is defined by the shape of the region `R⟨n,s,e,w⟩` × 5. The number 5 signifies the five slices of the cube, represents the declination angle, hour angle, latitude, xy-coordinate values, and digital surface values of each cell in `R⟨n,s,e,w⟩`. The temporal resolution of the overall tensor data-frame is defined by the day length (run rise to sunset with an equal time interval) for a given date.

### 3.2. Sun Position and Shadow-Top Coordinate Generation

TensorFlow [27] is an open-source software library developed by Google Brain Team. Nevertheless, it is known for mainly machine learning and deep learning; it is general enough to be applicable across various tasks in a wide variety of other domains. Tensorflow defines a general-purpose computational graph to achieve ease of expression. The Distributed Execution Engine of TensorFlow is compatible with GPUs and CPUs. The code

can be written using Python, C++, Java, and Go kind of frontend. The underlying Distributed Execution Engine then converts the code into hardware instruction sets. Tensors are the primary and central data structures TensorFlow uses as variables and feed in as placeholders into the computational graph to perform mathematical operations. These directed graphs with no recursion offer the implicit parallelism.

A TensorFlow core program comprises the following steps shown in Figure 8. A directed cyclic graph represents a model, where the nodes represent operations or primitive functions, and the edges the data flow between operations. Incident edges represent inputs into the node, and the edges incident out of a node represent the output. The data on edges can be either immutable tensors (`tf.constant`) or mutable variables (`tf.Variable`, `tf.placeholder`). Both tensors and variables are multi-dimensional arrays of primitive types. The execution of the computational graphs or part of the graphs can be initiated inside a session (`tf.Session`). It assigns resources (on one or more machines) and holds intermediate results and variables' actual values. When all their inputs are ready, the node gets ready to run. In the same way, multiple nodes can be executed in parallel when ready. The proof-of-concept implements Algorithms 1 and 2 defined in Sections 2.1 and 2.2, respectively using TensorFlow for its core computations and Python 3.6 for its front end. The underlying data-flow graph of the algorithms' implementation is shown in Figure 9 using TensorBoard. TensorFlow comes with an inbuilt interface, TensorBoard to visualize the data-flow graph and other tools to understand, debug, and optimize the model.

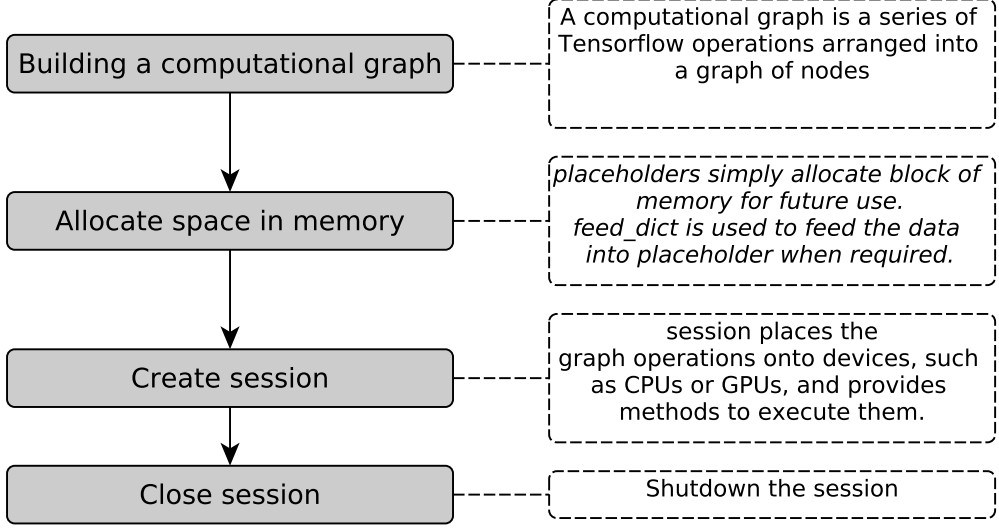

**Figure 8.** TensorFlow core program block diagram.

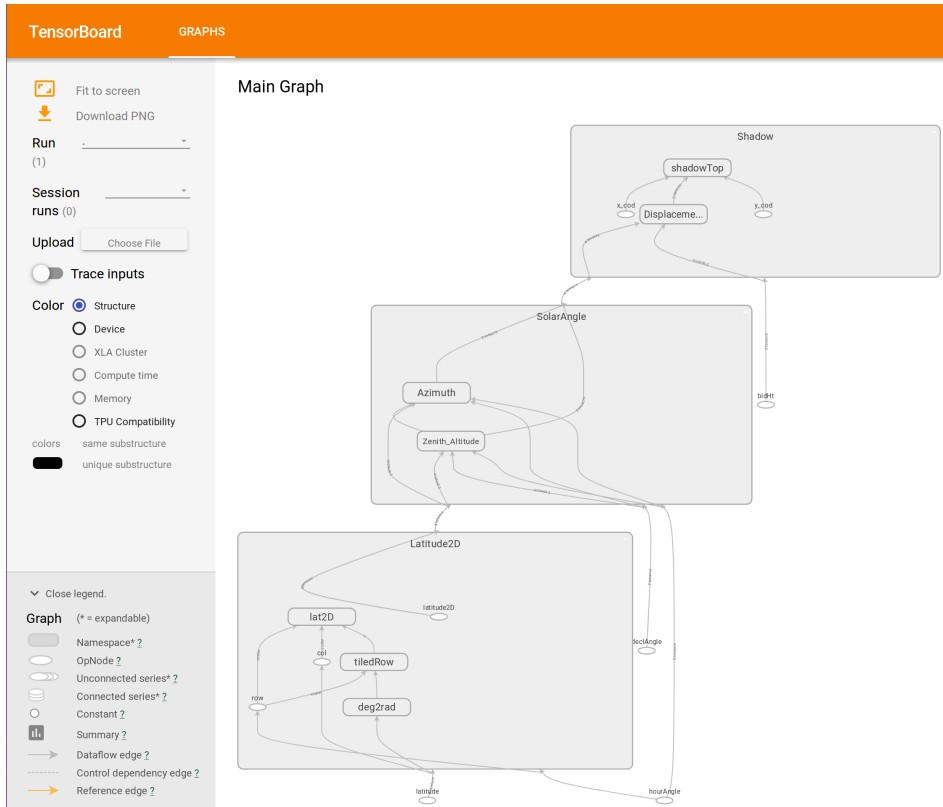

**Figure 9.** Dataflow graph generated by TensorBoard.

### 3.3. Shadow Point Detection

In 1962, Jack Elton Bresenham [30] developed an algorithm to draw lines on raster graphics devices. It is one of the fundamental line drawing algorithms in computer graphics. Given two points, the line-drawing algorithm identifies a set of points from an n-dimensional raster to constitute a close approximation to a straight line between the two points.

In Figure 10, the true line is the white straight line starting from (0,0) to (9,6), and the black cells along the true line are its approximation. The flowchart in Figure 11 illustrates Bresenham's line-drawing algorithm; first, the major and the minor axis are determined based on the length. For example, in Figure 10, X axis is the major axis. Starting from the original position, the current value of the major axis is incremented by exactly one cell (pixel) in each iteration. Then it scan for the most appropriate pixel on the minor axis for the current pixel on the major axis by checking individual pixels whose center is closer to the straight line. Thats it, Bresenham's algorithm all about this decision making. Although, it is not 100% precise but works adequately for higher resolution.

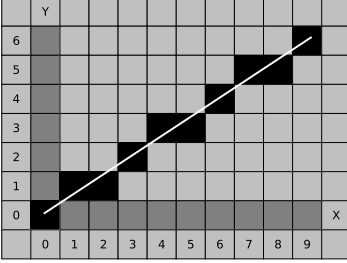

**Figure 10.** True line and its approximation.

In the proof-of-concept, Bresenham's line drawing algorithm is used to draw a line between the actual co-ordinates and associated shadow top co-ordinates derived by ex-

ecuting Algoritm 2 in Section 2.2. Naturally, for millions of such point-pair, applying Bresenham's algorithm is very time consuming and expensive. Accordingly, the algorithm is parallelized using PySpark. The obtained result exhibits high levels of scalability in terms of both time and accuracy.

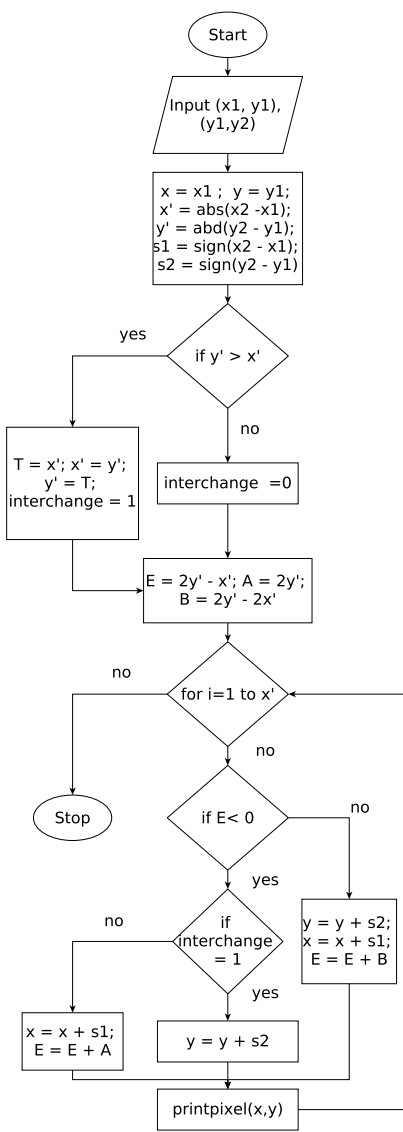

**Figure 11.** Bresenham's line drawing algorithm.

## 4. Experimental Results and Discussions

We select Esch-Sur-Alzette, the second-largest town after the capital Luxembourg city situated in the south-western Luxembourg border with France, as our test-bed. Esch-Sur-Alzette is situated at 49°29′44.988″ N, 5°58′50.016″ E, 5505.41 km north of the equator, in the northern hemisphere (https://fr.distance.to/Esch-sur-Alzette, accessed on 1 May 2020). It has a total area of 14.35 km$^2$ with elevation ranging from 279 m to 426 m. This Digital Surface Model is the result of a first LiDAR survey flight that has been done in October 2017 (https://data.public.lu/en/datasets/digital-surface-model-high-dem-resolution/, accessed on 1 May 2020). As a case study, the paper examines a region with upper-left and lower-right coordinates in EPSG:2169 (Luxembourg 1930/Gauss-https://epsg.io/2169, accessed on 1 May 2020) (66,838, 62,776) and (67,028, 62,640), respectively. The spatial resolution is 1 m. The associated DSM (mean sea level elevations)ranges between 285 m to 320 m including building, walls and trees. Shadows cast by structures vary in length and direction throughout the day and from season to season. The length of shadow from the

object is dependent on solar azimuth and solar altitude angle. Shadows are longer in the early morning and late afternoon. In the morning, the sun would rise from the east, and the shadow would be in the opposite direction (west), while in the evening, it would be in the reverse direction. Shadow lengths increase during the "low sun" or winter season and are longest on 21–22 December, the winter solstice. The winter solstice, therefore, represents the worst-case shadow condition, and the potential for loss of access to sunlight that a project could cause is highest. Shadow lengths are shortest on 21–22 June, the summer solstice. Both summer and winter solstice shadow patterns (drank burgundy colored) are examined in Figures 12 and 13, respectively. The Code focuses only on a given region R (Algorithm 3) and discards the rest beyond a specific boundary specified in R that is visible in Figures 12e,f and 13e,f. We execute our proof-of-concept on a 64-bit Intel(R) Core(TM) i7-6800K machine with a CPU @3.40GHz processor. Execution times for the shadow calculations are shown in Figure 14.

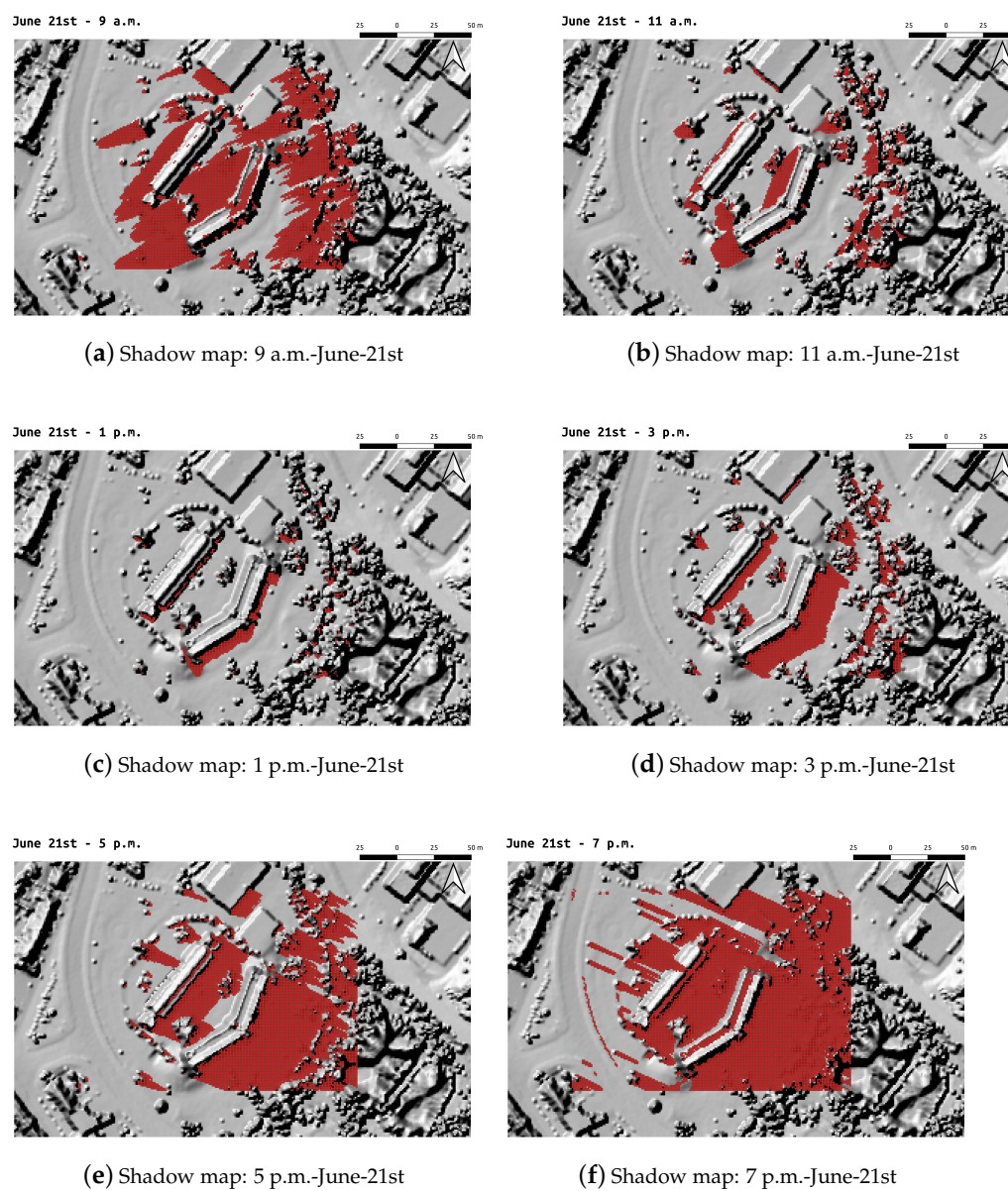

(**a**) Shadow map: 9 a.m.-June-21st

(**b**) Shadow map: 11 a.m.-June-21st

(**c**) Shadow map: 1 p.m.-June-21st

(**d**) Shadow map: 3 p.m.-June-21st

(**e**) Shadow map: 5 p.m.-June-21st

(**f**) Shadow map: 7 p.m.-June-21st

**Figure 12.** Shadow map for 21 June 2020 from 9 a.m. to 7 p.m. 2 h interval.

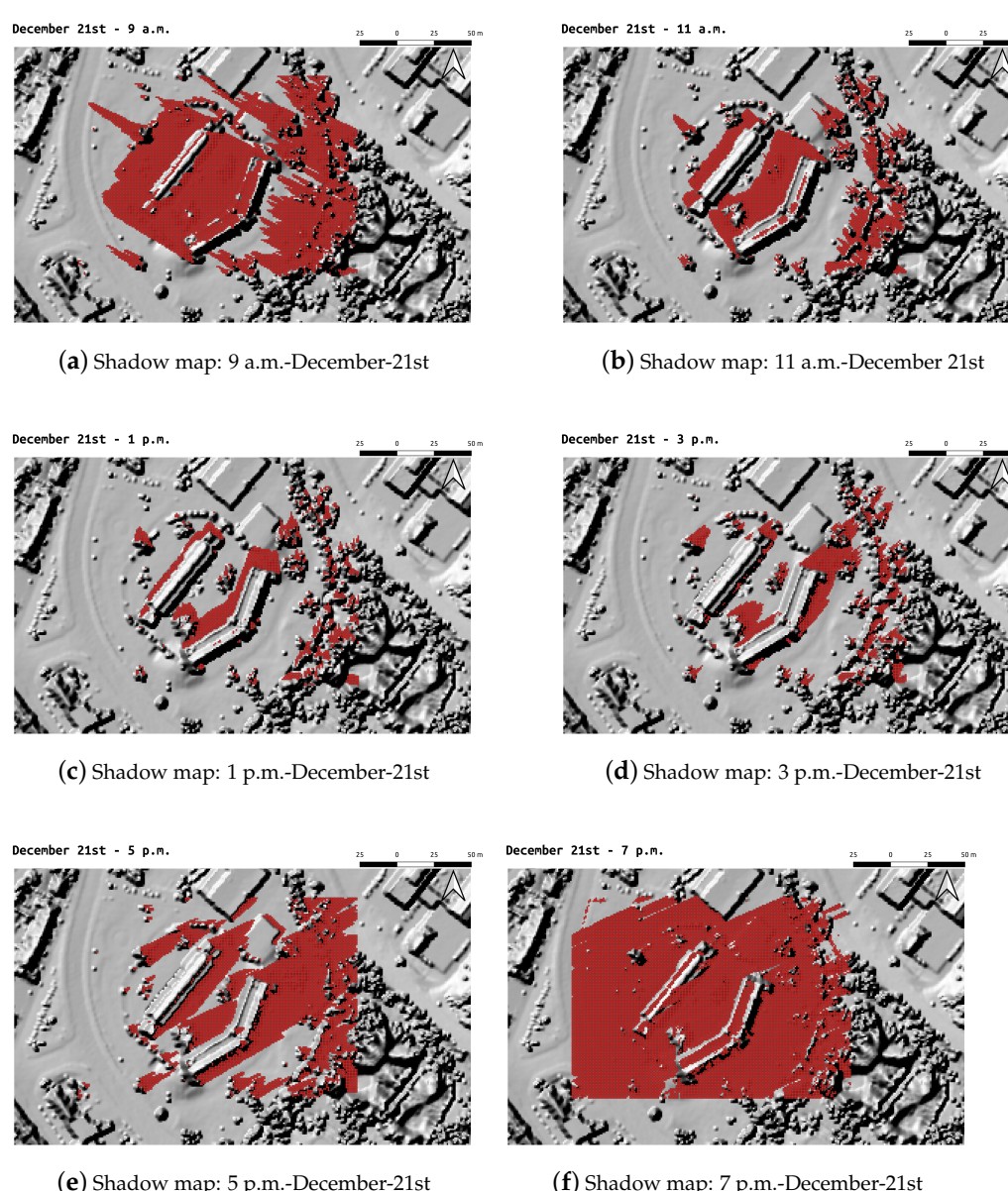

(**a**) Shadow map: 9 a.m.-December-21st

(**b**) Shadow map: 11 a.m.-December 21st

(**c**) Shadow map: 1 p.m.-December-21st

(**d**) Shadow map: 3 p.m.-December-21st

(**e**) Shadow map: 5 p.m.-December-21st

(**f**) Shadow map: 7 p.m.-December-21st

**Figure 13.** Shadow map for 21 December 2020 from 9 a.m. to 7 p.m. 2 h interval.

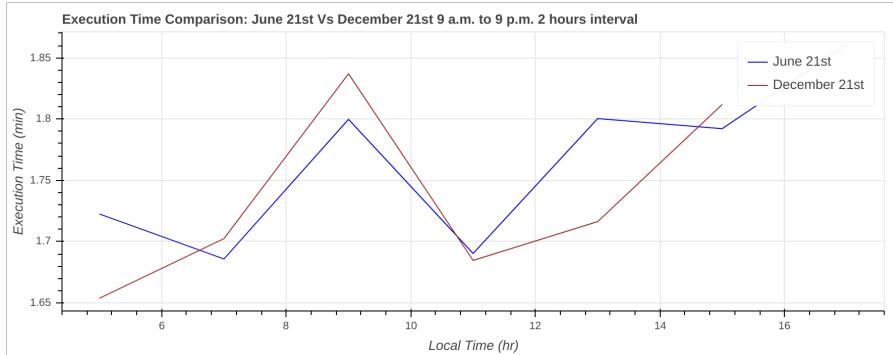

**Figure 14.** Execution time required for calculation.

## 5. Conclusions

Sustainable development is the key to urban planning and development. Researchers from various perspectives [32–34] investigate the requirement for shadow analysis by showing how solar access or shadow influences the city planning norms. Therefore, from the environmental perspective to city planning perspective or citizen's preferences, shadow analysis, more precisely, solar access, play a crucial role. Solar gain implies utility planning, building design, and overall city planning. Hence, smart cities should be planned to meet future energy demand and create an urban morphology to balance the urban net energy, reducing trapped heat into urban structures. A common impression given by the many shadow calculation algorithms examined is an expensive and time-consuming process. The "curse of dimensionality" addresses the problem caused by the exponential increase in volume associated with adding extra dimensions to Euclidean space [35]. Processing large scale multidimensional (spatio-temporal) data in raster-based geocomputation can cause the "curse of dimensionality" problem. We formulate such a situation into a tensor-based framework that explores different approaches to define a geospatial grid used to construct the tensor. Using the proposed framework, we developed a fast and accurate algorithm with theoretical guarantees for its convergence. We also validated the correctness and the efficiency of our implementation on real application datasets. The main characteristics of the proposed framework include

- Calculating mathematical expressions, in a more optimized and efficient manner, using tensors;
- Transparent use of GPU computing—CPU and GPU compatibility at the same time without changing the code;
- Underlying data-flow based implementation offers implicit parallelism and distributed execution with high scalability.
- GRASS GIS [17] 'Add-on' compatibility.

The proposed tensor-based framework also facilitates agile analytics on large-scale spatio-temporal data, including simulation, sensor data analysis, time-series analysis, and statistical data analysis.

**Author Contributions:** Conceptualization, Sukriti Bhattacharya; Data curation, Christian Braun; Formal analysis, Sukriti Bhattacharya; Funding acquisition, Ulrich Leopold; Investigation, Sukriti Bhattacharya; Methodology, Sukriti Bhattacharya, Christian Braun and Ulrich Leopold; Project administration, Ulrich Leopold; Software, Sukriti Bhattacharya; Supervision, Christian Braun and Ulrich Leopold; Validation, Sukriti Bhattacharya and Christian Braun; Visualization, Sukriti Bhattacharya; Writing – original draft, Sukriti Bhattacharya. All authors have read and agreed to the published version of the manuscript.

**Funding:** This work has been funded and supported by the ENOVOS Foundation Luxembourg and the Luxembourg Institute of Science and Technology (LIST) through the SECURE project.

**Institutional Review Board Statement:** Not applicable.

**Informed Consent Statement:** Not applicable.

**Data Availability Statement:** Sensitive and cannot be made publicly available.

**Conflicts of Interest:** The authors declare no conflict of interest. The funders had no role in the design of the study; in the collection, analyses, or interpretation of data; in the writing of the manuscript, or in the decision to publish the results.

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
