# Peer review of "An Efficient 2.5D Shadow Detection Algorithm for Urban Planning and Design Using a Tensor Based Approach"

_ijgi, doi:10.3390/ijgi10090583_

Round 1

Reviewer 1 Report

This paper proposes a tensor-based shadow computation algorithm for urban planning. There are several issues on this paper.

1 The outline of this paper is unclear and there are too many sections. I suggest the authors should rearrange this paper as follows:

(1) The section 2 “Sun Orientation” and section 3 “Shadow Geometry” can be combined into one section or added into a new section “Method” to show the minimal content since they are well known. Note the sub-section title 3.1 “shadow condition on DSM” should be deleted since there is only one subsection.  

(2) The content in section 4 “related work” can be moved to section 1 “introduction”.

(3) A new section “Method” can be added to summary all the algorithms used in this research. In its current form, the algorithms are scattered in all parts of the paper.

(4) A new section “Results and Discussion” can be added to show the results.

2 One of important words in the title is “real-time”. However, there is no discussion on this key performance and no definition of real-time.

3 There are no sufficient descriptions on the current research status. More related research should be added to justify the originality of this research.

4 The font in Figure 12 is too small to read.

5 In the application of the method proposed in this paper, the authors only show the results from this method. The results should be compared with the other more mature methods to validate the results and compare the computational time.

6 Figure 7 may be deleted since there is no sufficient information to be delievered.

Author Response

Dear reviewer,
Thank you so much for your insightful comments on our paper. We have been able to incorporate changes to reflect most of the suggestions provided by the reviewers. We have highlighted the changes within the manuscript. Here is a point-by-point response to the reviewers' comments and concerns.

Comment (1) and (3): Section 2, "Sun Orientation," and Section 3, "Shadow Geometry," are combined into one section, "Methodology." Individual sections become subsections under Section 2: Methodology, and this way, all the necessary algorithms are under one section, not scattered like before.
Comment (2):  The content of Section 4, "Related Work," is added to Section 1, "introduction." 
Comment(4): i) Section 4, "Experimental Results and Discussions," are added.
ii) The title is improved as we remove the real-time work, which was indeed not appropriate with the context.
iii) related work added.
iv) Figure 7 is removed.

We tried our best to improve the manuscript and made some changes in the manuscript. We appreciate for reviewer's warm work earnestly and hope that the correction will meet with approval. Once again, thank you very much for your comments and suggestions.

Sincerely, 

Dr. Sukriti Bhattacharya

Reviewer 2 Report

The paper presents an interesting approach using tensors to calculate the hard shadow. The approach for the shadow algorithm is not really new, it has been published, with the difference that the name tensors was not used, multiple matrices were used instead, all related by the row and column indices. Parallel computing for the shadow algorithm was also already used. The execution times are also not very much smaller than in other methods, taking into account the small dimensions of the calculated area. The method seems to be only suitable for shadows on the ground, not on facades, either cast or self shadows. Nevertheless, since it is a somewhat different computation approach, using tensorflow, it's worthy to be explored. Some sections are full of definition errors. The chosen figures for explaining the sun coordinates are not the best. There seem to be some problems in the algorithms, since the resulting shadow maps do not seem logic, especially for July 21st. Please check. Some sections are not fluently explained. They include too much code expressions for a normal reader. For the rest, please read the attached commented pdf file.

Author Response

Dear reviewer,
Thank you so much for your insightful comments on our paper. We have been able to incorporate changes to reflect most of the suggestions provided by the reviewers. We have highlighted the changes within the manuscript. Here is a point-by-point response to the reviewers' comments and concerns.

1) Figure 2 is replaced by a more convincing and understandable one and explained in Section 2.1 that explained solar coordinates and angles.

2) Figure 12 is explained in Section 4, "Experimental Results and Discussions."

3) each and every minor correction made in this document "peer-review-12987375.v1.pdf" is refected.

We tried our best to improve the manuscript and made some changes in the manuscript. We appreciate for reviewer's warm work earnestly and hope that the correction will meet with approval. Once again, thank you very much for your comments and suggestions.

Sincerely,

Dr. Sukriti Bhattacharya

Reviewer 3 Report

  • The introduction does't justify the scope of paper clearly. What 2.5D means? Why 2.5D? what's the benefit compared to existing methods or dimensions (1D or 3D)? it looks vague without enough background & related references; in Section 4 mentioned that raster-based solutions are widely implemented, so what is the efficiency/benefit here? how this paper is different. 
  • Figure 2 diagram is difficult to understand, is it a plan view or section?
  • Good to provide a structure for the methodological steps
  • Did you validate the results?

Author Response

Dear reviewer,
Thank you so much for your insightful comments on our paper. We have been able to incorporate changes to reflect most of the suggestions provided by the reviewers. We have highlighted the changes within the manuscript. Here is a point-by-point response to the reviewers' comments and concerns.

1) Comment on 2.5D: The concept is explained in Section 1, "Introduction," and a related paper is also added [24] for better understanding.

2) Comment on Figure 2:  Figure is replaced by a more convincing and understandable one and explained in Section 2.1.

3) Pictorial representation of the whole methodology: Figure 11, which is explained in Section 3, "Proof-of-Concept."

We tried our best to improve the manuscript and made some changes in the manuscript. We appreciate for reviewer's warm work earnestly and hope that the correction will meet with approval. Once again, thank you very much for your comments and suggestions.

Sincerely, 

Dr. Sukriti Bhattacharya

Round 2

Reviewer 1 Report

The authors have addressed all the issues.